# MoEUT: Mixture-of-Experts Universal Transformers

**Róbert Csordás**[1,2†]    **Kazuki Irie**[3]    **Jürgen Schmidhuber**[2,4]
**Christopher Potts**[1]    **Christopher D. Manning**[1]
[1]Stanford University, Stanford, CA, USA
[2]The Swiss AI Lab IDSIA, USI & SUPSI, Lugano, Switzerland
[3]Center for Brain Science, Harvard University, Cambridge, MA, USA
[4]AI Initiative, KAUST, Thuwal, Saudi Arabia
`{rcsordas,cgpotts,manning}@stanford.edu`
`kirie@fas.harvard.edu, juergen@idsia.ch`

## Abstract

Previous work on Universal Transformers (UTs) has demonstrated the importance of parameter sharing across layers. By allowing recurrence in depth, UTs have advantages over standard Transformers in learning compositional generalizations, but layer-sharing comes with a practical limitation of parameter-compute ratio: it drastically reduces the parameter count compared to the non-shared model with the same dimensionality. Naively scaling up the layer size to compensate for the loss of parameters makes its computational resource requirements prohibitive. In practice, no previous work has succeeded in proposing a shared-layer Transformer design that is competitive in parameter count-dominated tasks such as language modeling. Here we propose MoEUT (pronounced "moot"), an effective mixture-of-experts (MoE)-based shared-layer Transformer architecture, which combines several recent advances in MoEs for both feedforward and attention layers of standard Transformers together with novel layer-normalization and grouping schemes that are specific and crucial to UTs. The resulting UT model, for the first time, slightly outperforms standard Transformers on language modeling tasks such as BLiMP and PIQA, while using significantly less compute and memory.[1]

## 1 Introduction

Transformers [1, 2] are ubiquitous neural architectures in modern machine learning. They power large language models [3, 4, 5, 6, 7], modern image processors [8], offline reinforcement learning agents [9], and many others. Despite these successes, we should ask whether more optimal architectures exist.

One important candidate is the Universal Transformer (UT, [10]). The core characteristic of UTs is *recurrence in depth* via *sharing parameters across layers*. This reintroduces the expressive power of recurrence provided by recurrent neural networks (RNNs, [11, 12, 13]). Layer sharing allows UTs to outperform regular Transformers on compositional problems such as logical inference tasks, while also yielding improvements on *small-scale* language modeling and translation tasks. In particular, UTs have been shown to have better compositional generalization properties [14, 15] by being able to decompose structured problems without supervision and generalize to longer sequences [16].[2] These empirical findings confirm that UTs are more general architectures with superior

---

[†]Work started at IDSIA.

[1]Our code is public: `https://github.com/robertcsordas/moeut`

[2]Dehghani et al. [10] also augment UTs with an additional adaptive computation time (ACT, [17, 18]) mechanism. However, the benefits of UTs we discuss here are purely due to layer-sharing, which, in consequence, is the focus of this work. Our models could also optionally be augmented with ACT but this is out of scope here.

38th Conference on Neural Information Processing Systems (NeurIPS 2024).

generalization properties compared to standard Transformers, in principle. However, UTs suffer from a fundamental problem of *parameter–compute ratio*: sharing the parameters among $L$ layers of an $L$-layer Transformer—while keeping the same model dimensionalities—results in a model with $L$ times fewer parameters (ignoring the input/output layers to simplify the discussion). Upscaling the size of the layer to compensate for the loss of parameters (essentially by making it $L$ times wider) usually yields a very big layer whose computational requirements in terms of compute and memory are prohibitive in practice [19, 20]. In sum, despite their potential, UTs are much less compute-efficient than standard Transformers, and thus, they are not popular for parameter-dominated tasks such as modern language modeling. Indeed, we are not aware of any previous work that has succeeded in developing compute-efficient UT models that yield competitive performance compared to standard Transformers on such tasks.

Here we bring new perspectives and a solution to UTs' fundamental compute–parameter ratio problem. We present Mixture-of-Experts Universal Transformers (MoEUTs, pronounced "moot"), a mixture-of-experts (MoE) architecture [21, 22, 23] for UTs enabling them to scale in a computationally and memory efficient way. We leverage various recent advances in MoEs for both feedforward and self-attention layers (Sec. 2.1 and 2.2), and combine them with two new innovations: (1) *layer grouping*, in which we recurrently stack groups of MoE-based layers, and (2) a *peri-layernorm* scheme (which is "in-between" the standard pre- and post-layernorm), in which we apply layer norm only before linear layers that immediately precede sigmoid or softmax activations. Both are specifically designed for shared-layer MoE architectures, and strongly supported by empirical evidence.

MoEUTs allow us to build parameter- and resource-efficient UT language models outperforming standard Transformers with less compute and memory requirements on all scales on which we can afford to test (up to 1B parameters). We demonstrate their capabilities on the C4, SlimPajama, and peS2o language modeling datasets, as well as on The Stack code generation. Our experiments show that recurrence is essential for our models to achieve competitive performance. We also demonstrate good zero-shot performance on downstream tasks like BLiMP and Children's Book Test, Lambada, HellaSwag, PIQA and ARC-E.

## 2 The MoEUT Architecture

Our MoEUT architecture is a Transformer architecture with shared layer parameters, in which we address the parameter-compute ratio problem by using mixture-of-experts. While there are many recent works on MoE methods for Transformer language models (e.g., [24, 25, 26, 27, 28]), making them competitive against their dense counterparts in *parameter-equal* comparisons is known to be challenging [28]. Here we leverage recent advances in MoE methods for both the feedforward network block (FFN, or simply MLP layer or feedforward layer; Sec. 2.1) and the self-attention layer (Sec. 2.2) together with two novel methods that take into account the specific properties of shared-layer models, namely: layer grouping (Sec. 2.3) and signal propagation (Sec. 2.4), which, taken together, are crucial for achieving effective shared-layer MoE Transformers.

### 2.1 MoE Feedforward Blocks

To parameterize the feedforward blocks of our shared-layer Transformers by an MoE, we use $\sigma$-MoE [28] with a few modifications. $\sigma$-MoE divides the feedforward block into $N_E$ slices, called *experts*. Each expert has two sets of weights, $\boldsymbol{W}_1^e \in \mathbb{R}^{d_{\text{model}} \times d_{\text{expert}}}$ and $\boldsymbol{W}_2^e \in \mathbb{R}^{d_{\text{expert}} \times d_{\text{model}}}$, where $e \in \{1, \dots, N_E\}$ is the index of the expert. At each token position $t$, given layer input $\boldsymbol{x}_t \in \mathbb{R}^{d_{\text{model}}}$, the MoE feedforward layer computes a score for each expert, yielding a vector $\boldsymbol{s} \in \mathbb{R}^{N_E}$ computed as:

$$\boldsymbol{s}_t = \sigma(\boldsymbol{x}_t \boldsymbol{W}_S) \tag{1}$$

where $\boldsymbol{W}_S \in \mathbb{R}^{d_{\text{model}} \times N_E}$ is a trainable weight matrix, and $\sigma(x) = \frac{1}{1+e^{-x}}$ is the element-wise sigmoid function. The MoE layer only selects $K$ experts (out of $N_E$) corresponding to the top-$K$ elements in $\boldsymbol{s}_t \in \mathbb{R}^{N_E}$ to produce the layer output $\boldsymbol{y}_t \in \mathbb{R}^{d_{\text{model}}}$ as follows:

$$\mathcal{E}(\boldsymbol{x}_t) = \arg \operatorname{topk}(\boldsymbol{s}_t, K) \subseteq \{1, \dots, N_E\} \tag{2}$$

$$\boldsymbol{y}_t = \sum_{e \in \mathcal{E}(\boldsymbol{x}_t)} \boldsymbol{s}_t[e] \operatorname{ReLU}(\boldsymbol{x}_t \boldsymbol{W}_1^e) \boldsymbol{W}_2^e \tag{3}$$

where $\boldsymbol{s}_t[e] \in \mathbb{R}$ is the $e$-th element of vector $\boldsymbol{s}_t \in \mathbb{R}^{N_E}$. Our preliminary experiments revealed that the original regularization of $\sigma$-MoE tends to be unstable and sometimes causes loss explosion during training. To avoid this, we apply regularization only within the sequence (as opposed to all tokens in the batch). For a sequence of inputs $\boldsymbol{x}_t, t \in \{1, \ldots, T\}$ we compute the balancing loss $L$ as:

$$L = \sum_{e=1}^{N_E} \boldsymbol{p}[e] \log \boldsymbol{p}[e], \qquad \boldsymbol{p} = \frac{1}{T} \sum_{t=1}^{T} \text{softmax}(\boldsymbol{x}_t \boldsymbol{W}_S) \in \mathbb{R}^{N_E} \tag{4}$$

The loss is scaled with coefficient $\gamma$ and added to the standard cross entropy loss. Unlike the original $\sigma$-MoE, no expert dropout is used in our experiments. It is important to note that, in contrast to the standard setup in the MoE literature, our experts are small ($d_{\text{expert}} = 128$, similarly to $\sigma$-MoE [28]), and there are 100s of them. This configuration is called *fine-grained* mixture-of-experts [29] and is also advocated by Dai et al. [30]. We analyze the effect of $d_{\text{expert}}$ in Fig. 13 in the appendix.

## 2.2 MoE Self-Attention Layers

To introduce MoE to the self-attention layers, we apply SwitchHead [31], which is an MoE method extending $\sigma$-MoE to attention layers. As in the standard multi-head attention layer, each head in the SwitchHead layer contains four transformations: query, key, value, and output projections. However, SwitchHead parameterizes the value and output projections using MoEs. That is, each head has one query and key projection associated with it and $N_A$ value and output projections, which are chosen dynamically for each input. Keys and queries are computed "as usual": given an input at position $t$, $\boldsymbol{x}_t \in \mathbb{R}^{d_{\text{model}}}$, $\boldsymbol{k}_t^h = \boldsymbol{x}_t \boldsymbol{W}_K^h$ and $\boldsymbol{q}_t^h = \boldsymbol{x}_t \boldsymbol{W}_Q^h$, where $\boldsymbol{W}_K^h$ and $\boldsymbol{W}_Q^h \in \mathbb{R}^{d_{\text{model}} \times d_{\text{head}}}$, where $h \in \{1, \ldots, H\}$ is the head index. The expert selection for the values is computed as follows:

$$\boldsymbol{s}_{V,t}^h = \sigma(\boldsymbol{x}_t \boldsymbol{W}_{SV}^h) \in \mathbb{R}^{N_A} \tag{5}$$

$$\mathcal{E}_V^h(\boldsymbol{x}_t) = \arg \text{topk}(\boldsymbol{s}_{V,t}^h, K_A) \subseteq \{1, \ldots, N_A\} \tag{6}$$

where $\boldsymbol{W}_{SV}^h \in \mathbb{R}^{d_{\text{model}} \times N_A}$ is the selection weight for the value and $K_A$ is the number of simultaneously active experts per head, set to $K_A = 2$ in all of our experiments. The selection for the values and outputs are independent. The selection of the output is computed analogously using a different weight matrix $\boldsymbol{W}_{SO}^h \in \mathbb{R}^{d_{\text{model}} \times N_A}$: $\boldsymbol{s}_{O,t}^h = \sigma(\boldsymbol{x}_t \boldsymbol{W}_{SO}^h) \in \mathbb{R}^{N_A}$ and $\mathcal{E}_O^h(\boldsymbol{x}_t) = \arg \text{topk}(\boldsymbol{s}_{O,t}^h, K_A) \subset \{1, \ldots, N_A\}$. Then the output $\boldsymbol{y} \in \mathbb{R}^{d_{\text{model}}}$ is calculated as follows:

$$\boldsymbol{v}_t^h = \sum_{e \in \mathcal{E}_V^h(\boldsymbol{x}_t)} \boldsymbol{s}_{V,t}^h[e] \boldsymbol{x}_t \boldsymbol{W}_V^{h,e} \in \mathbb{R}^{d_{\text{head}}} \tag{7}$$

$$\boldsymbol{a}_t^h = \text{Attention}(\boldsymbol{q}_t^h, \boldsymbol{K}_t^h, \boldsymbol{V}_t^h) \in \mathbb{R}^T \tag{8}$$

$$\boldsymbol{y}_t = \sum_{h=1}^{H} \sum_{e \in \mathcal{E}_O^h(\boldsymbol{x}_t)} \boldsymbol{s}_{O,t}^h[e] \boldsymbol{a}_t^h \boldsymbol{V}_t^h \boldsymbol{W}_O^{h,e} \tag{9}$$

where $\boldsymbol{W}_V^{h,e} \in \mathbb{R}^{d_{\text{model}} \times d_{\text{head}}}$ and $\boldsymbol{W}_O^{h,e} \in \mathbb{R}^{d_{\text{head}} \times d_{\text{model}}}$ are head $h$, expert $e$, weight matrices for value and output respectively; $\boldsymbol{s}_{V,t}^h[e], \boldsymbol{s}_{O,t}^h[e] \in \mathbb{R}$ are scores of expert $e$ for head $h$ at position $t$ for value and output MoE respectively; and $\text{Attention}$ denotes the standard softmax scaled dot attention [1] with $\boldsymbol{K}_t^h = (\boldsymbol{k}_1^h, \ldots, \boldsymbol{k}_t^h), \boldsymbol{V}_t^h = (\boldsymbol{v}_1^h, \ldots, \boldsymbol{v}_t^h) \in \mathbb{R}^{T \times d_{\text{head}}}$ e.g., for the auto-regressive setting. Note that, here we describe position-wise computations for clarity but in practice, they can be parallelized over the tokens through matrix operations. Unlike in the original SwitchHead, which uses no regularization, we apply the same entropy regularization we use in the feedforward layer (Eq. 4) with a regularization coefficient $\delta$. (The same value is used for both value and output.)

## 2.3 Layer Grouping: MoE-efficient Layer Sharing & Sub-operations within an Operation

Even when using the two recent MoE methods above, which have been shown to be successful for the standard Transformer (Sec. 2.1 and 2.2), we experimentally observe that naive MoE-based UTs with a *single* shared layer often struggle to achieve good performance at larger scales. We hypothesize that the reason is twofold. First, as the network scales, the number of experts in the layer grows rapidly, but we cannot increase the number of active experts $K$ at the same rate without greatly increasing the required compute. This forces us to reduce the percentage of active experts, which is generally

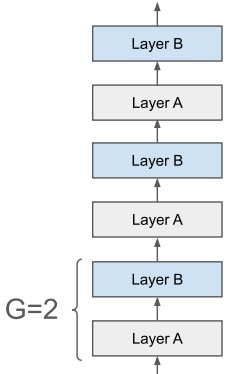
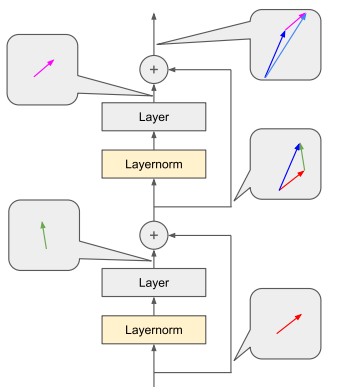
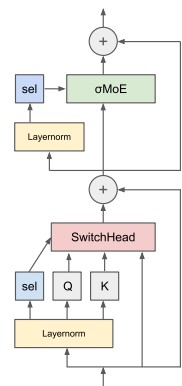

Figure 1: Layer grouping: 8 layers with group size of 2.

Figure 2: The residual grows in pre-layernorm transformers.

Figure 3: MoEUT block with no layernorms in the residual.

detrimental. Second, the total number of attention heads is kept relatively low, which might not be sufficient for a large model. Increasing their number is similarly prohibitively expensive.

Our solution to these problems is to stack multiple layers with *non-shared* weights to form what we call a *group* of layers, reducing the number of experts in each $\sigma$-MoE while increasing the total number of attention heads. The final network is obtained by recurrently stacking such *groups* that share the same parameters (in a sense, redefining the *group* as a shared "layer" in the UT). Fig. 1 provides an illustration; here, all layers denoted by "Layer A" (or "B" respectively) share the same parameters across the entire network. The size of the group, $G$, is the number of non-shared layers in it. In our experiments, the group size is between 2 and 4, and the typical number of recurrent steps is 8 or 9.

As further observations in favor of the potential inductive bias introduced by such grouping, note that in a seminal work, Olsson et al. [32] reverse engineer one of the main mechanisms behind in-context learning: induction heads. They find that two successive layers where the attention performs different operations in each layer are required. Furthermore, Csordás et al. [16] also show that their shared-layer Transformers use two consecutive layers to perform a *single* operation for relatively complex synthetic tasks, such as ListOps. Both of these observations indicate that the adjacent layers in Transformers often perform different sub-operations for a *single* high-level step of computation that spans multiple layers. This is well aligned with our proposed grouping.

## 2.4 Novel LayerNorm Scheme for Improved Signal Propagation in Universal Transformers

Virtually all modern Transformers make use of the so-called "pre-layernorm" scheme [33, 34] (as opposed to the "post-layernorm" one), that is, layer normalization [35] is applied before the attention layer (or analogously, the feedforward block), and their output is directly added to the residual. The residual is normalized only before the final classification layer. This design encourages better gradient flow and is often crucial for training deep models. This indicates that the norm of the residual vector should grow as we go deeper in the network (see Fig. 2 for an illustration). However, it is typically assumed that the information is carried in the *direction* of the residual vector instead of its length [36, 37]. Because of this, late layers must learn to produce outputs with a larger norm so that they can apply the same order of modification to the residual as the earlier ones, despite having normalized inputs because of the layernorm.

These learning targets are easily achieved by standard Transformers, as they have separate parameters which can have different scalings in different layers, and this can be observed empirically (for more details, see Appendix A.2). This is not the case for UTs as they have a single, shared layer (or in our case multiple, repeated layers; see Sec. 2.3). If some circuits should be (re-)used in both early and late layers, scaling their output to compensate for the norm growth of the residual is nontrivial.

Post-layernorm does not have this problem, since the whole residual is normalized after each layer. This coincides with the observation of Tan et al. [38] that post-layernorm performs better for UTs than pre-layernorm, and with the fact that the original UT [10] is trained with post-layernorm. That said, as mentioned above, post-layernorm also has its own limitation in terms of gradient flow [34].

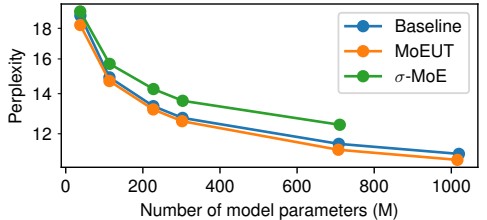
(a) Scaling in the number of parameters.

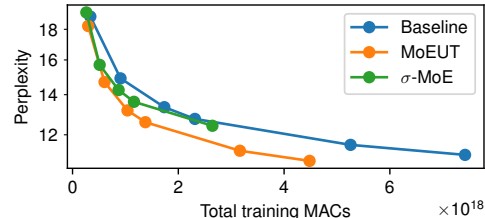
(b) Scaling in the number of MACs used for training.

Figure 4: Scaling of different models on C4 (with perplexity measured on a held-out subset of C4). (a) MoEUT slightly outperforms parameter-matched models with no layer sharing. The gap grows with scale. (b) Given equal amounts of compute, MoEUT outperforms other models by a large margin.

Here we propose an alternative method to avoid the aforementioned problems: we do not use layernorms in the "main data path". This means, for our UTs, that we apply no layernorm before the value projection of the attention and no layernorm before the $\sigma$-MoE layer. Rather, layernorm is used only before linear layers that are immediately followed by a sigmoid or softmax activation function (producing renormalized activations that are critical before these nonlinear layers), namely: the query and key projections in the attention, the expert selection on both the attention and feedforward layers, and before the final classification layer. This is illustrated in Fig. 3. Since only a ReLU activation function is used on the main data path inside the feedforward layer, the output updates will be proportional to the input, thus effectively solving the residual growth issue while also providing efficient gradient flow paths. We call this the "peri-layernorm" scheme as a scheme "between" pre- and post-layernorm, which positions layernorm "around" (but not on) the residual connections.

## 3  Main Experimental Results

We present our main experimental results on the performance and efficiency of MoEUT on language modeling using the popular C4 dataset [39]. To demonstrate the versatility of our model, we also show our main results on the SlimPajama [40] and peS2o [41] language modeling datasets, and code generation on "The Stack" [42]. For experimental evidence in support of the benefits of shared layers for compositional generalization, we refer to much previous work (e.g., [10, 15, 14, 16, 38]). Following prior work [27, 31], we measure the compute requirements in terms of the number of multiply-accumulate (MAC) operations needed in the forward pass.

Because our models are fully MoE, they decouple the number of parameters, compute and memory requirements, and different model dimensions such as $d_{\text{model}}$ and $d_{\text{ff}}$, number of layers. Thus, they provide greater flexibility for model designers. We follow a simple procedure for setting the model's hyperparameters, as described below. All our models use RoPE positional encodings [43] with PyTorch's fast attention implementation. The baseline models are pre-layernorm Transformers. For each baseline, we construct a *parameter-matched* MoEUT model. We set $d_{\text{model}}$ and the number of layers $n_{\text{layers}}$ to be the same as for the dense baseline. We use the same tokenization for each model trained on the same dataset. The number of heads $H$ for MoEUT is set to $\frac{1}{4}H$ of the corresponding dense model, $d_{\text{head}}$ is set to $2d_{\text{head}}$ of the corresponding dense model, and we set $K_A = 2$. This matches the number of MACs spent for the value and output projections in self-attention, and reduces the number of MACs spent on calculating keys and queries and the attention matrices itself. For the $\sigma$-MoE layers, we set the expert size $d_{\text{expert}} = 128$, and $K = 2d_{\text{model}}/d_{\text{expert}}$. This halves the MAC requirements compared to the dense counterpart. We set the number of experts in the feedforward block, $N_E$, and the number of attention experts, $N_A$ such that the number of parameters is the same as for the dense baseline, and $10-15\%$ of the model's parameter budget (excluding the embedding and classification layers) is spent in the attention computations. We set the group size $G$ to 2 for all our models below 300M parameters, $G = 3$ for our 319M parameter model, and $G = 4$ for the bigger models. This helps keep the number of experts manageable and improves both the performance and the speed of the model. All models are trained with batch size 64 and context length 1024, for $10^5$ steps. This protocol allows us to perform fair comparisons between different models within our

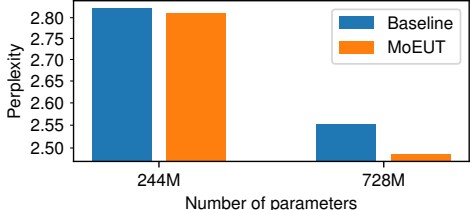
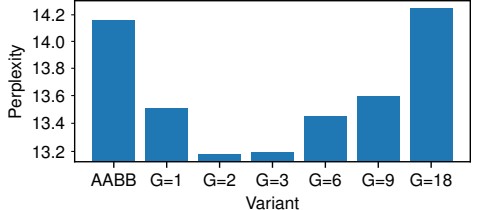

Figure 5: Performance of MoEUT compared to a standard Transformer on **The Stack**. MoEUT outperforms standard Transformers. The gap grows with scale.

Figure 6: Perplexity of 244M MoEUT models with different layer grouping options. A small group size of $G = 2$ works the best, showing the advantage of layer sharing.

computational budget, and it leads to high quality models, as measured by our benchmarks. For more details, see Appendix A.4.

**Scaling compared to standard Transformers.** Our main scaling results are shown in Fig. 4. The y-axis shows the perplexity on a held-out subset of C4. The plot shows that our MoEUT model slightly outperforms dense models with the same number of parameters (Fig. 4a), and the gap tends to grow with scale. Additionally, we compare to the non-shared $\sigma$-MoE model [28]. This $\sigma$-MoE baseline has the same shape of feedforward layers ($d_{model}$, $K$, $d_{expert}$) to our layer-shared MoEUT, but uses no attention experts to keep the proportion of the attention weights as close MoEUT as possible, and it also uses our peri-layernorm scheme (Sec. 2.4). We add this baseline as the model that is as close to our shared-layer model as possible. This model performs significantly worse than MoEUT, demonstrating the clear advantage of the shared layers. Additionally, Fig. 4b shows that in terms of the number of total MAC operations spent on all forward passes during training, MoEUT outperforms the baseline dense model by a large margin.

**Performance on code generation.** To confirm the effectiveness of our model on a different task domain, here we train it on a subset of the "The Stack" dataset [42] which is a code generation task. As we cannot afford a full epoch of training, we limit ourselves to a few languages only. We use a mixture of diverse languages: Python, HTML, C++, Rust, JavaScript, Haskell, Scala, and assembly. We evaluate our models on a held-out subset of the dataset. The results are shown in Fig. 5, and they are in line with our findings on the natural language domain: MoEUT outperforms the baseline.

**Zero-shot performance on downstream tasks.** Here we evaluate the zero-shot performance of our models on six different downstream tasks: LAMBADA [44], BLiMP [45], Children's Book Test (CBT) [46], HellaSwag [47], PIQA [48], and ARC-E [49]. For LAMBADA, we use the detokenized version from OpenAI, and we evaluate the top-1 accuracy of the last word (it can span multiple tokens; here we use greedy decoding). For CBT and BLiMP, we measure the accuracy for each task and report the average of the tasks' accuracies. The results are shown in Tab. 1. We observe that our models and the baselines typically perform very similarly. MoEUT often outperforms the baseline, but the differences are marginal in all cases. This confirms that our models are indeed very capable compared to standard language models. We confirm this on peS2o and SlimPajama as well.

**Comparing with SUT.** Here we compare our MoEUT to another baseline, Sparse Universal Transformer (SUT; [38]), which is a recently proposed UT model that also makes use of MoE layers. We note that SUTs have not been evaluated previously on standard language modeling tasks. While both MoEUT and SUT make use of an MoE for both feedforward and attention layers, there are several technical differences at various levels between the two methods: SUT uses competitive expert selection (softmax), multiple load balancing losses, and much bigger expert sizes. Their model is post-layernorm and does not use layer grouping. Unlike ours, Adaptive Computation Time (ACT) is used in the layer dimension.

We took the original code released by Tan et al. [38] and ported it to our training pipeline for a fair comparison. As for MoEUT, we roughly match the model's dimensionalities and number of active channels to our dense baselines. We ran a hyperparameter optimization for the regularization losses,

Table 1: Zero-shot downstream performance and perplexity on various language modeling datasets. MoEUT marginally outperforms standard Transformers in most tasks, confirming that MoEUT is indeed a capable language model.

| Dataset | #params | Model | PPL ↓ | LAMBADA ↑ | BLiMP ↑ | CBT ↑ | HellaSwag ↑ | PIQA ↑ | ARC-E ↑ | Average ↑ |
|---|---|---|---|---|---|---|---|---|---|---|
| C4 | 44M | Baseline | 18.97 | 21.9% | 73.5% | 81.3% | 28.3% | 59.9% | 31.7% | 49.4% |
| | | MoEUT | **18.30** | **23.2%** | **78.2%** | 81.1% | **29.2%** | **61.3%** | **33.5%** | **51.1%** |
| | 126M | Baseline | 14.97 | **28.5%** | 77.0% | **84.4%** | 31.7% | 62.7% | 35.2% | 53.2% |
| | | MoEUT | **14.76** | 27.2% | **79.4%** | 84.2% | **32.3%** | **64.4%** | **35.3%** | **53.8%** |
| | 244M | Baseline | 13.40 | **33.1%** | 78.5% | **86.0%** | 34.5% | 64.9% | **36.9%** | **55.6%** |
| | | MoEUT | **13.24** | 30.6% | **79.7%** | 85.3% | **35.7%** | **65.2%** | 36.4% | 55.5% |
| | 319M | Baseline | 12.81 | **33.3%** | 78.5% | **87.2%** | 36.1% | **67.1%** | 37.2% | 56.6% |
| | | MoEUT | **12.65** | 30.8% | **80.2%** | 86.9% | **37.3%** | 67.0% | **37.3%** | 56.6% |
| | 728M | Baseline | 11.59 | **37.8%** | 80.7% | 88.2% | 40.5% | 67.7% | 39.3% | 59.0% |
| | | MoEUT | **11.34** | 36.0% | **80.8%** | **88.4%** | **41.8%** | **69.2%** | **39.6%** | **59.3%** |
| | 1040M | Baseline | 11.15 | **38.4%** | 81.2% | 89.0% | 42.0% | 68.6% | 39.7% | 59.8% |
| | | MoEUT | **10.90** | **38.4%** | **81.6%** | **89.2%** | **43.7%** | **69.9%** | **41.3%** | **60.7%** |
| peS2o | 44M | Baseline | 11.46 | **13.2%** | 66.5% | 68.6% | **28.5%** | **56.3%** | **32.0%** | 44.2% |
| | | MoEUT | **11.09** | 13.1% | **68.7%** | **69.6%** | 28.3% | 55.1% | 31.4% | **44.4%** |
| | 244M | Baseline | 8.55 | 18.7% | 72.8% | **78.0%** | **30.4%** | **56.3%** | 35.0% | 48.5% |
| | | MoEUT | **8.52** | **19.4%** | **73.5%** | 77.4% | 30.1% | **56.3%** | **35.6%** | **48.7%** |
| SlimPajama | 44M | Baseline | 16.42 | **20.0%** | 72.8% | 80.7% | 27.5% | 57.0% | 31.6% | 48.3% |
| | | MoEUT | **15.77** | 19.8% | **75.9%** | **82.1%** | **28.0%** | **57.5%** | **32.1%** | **49.2%** |
| | 244M | Baseline | 11.51 | **31.9%** | 78.6% | **87.3%** | 31.7% | 60.9% | **36.6%** | **54.5%** |
| | | MoEUT | **11.47** | 30.7% | **80.2%** | 86.8% | **32.0%** | **61.7%** | 35.8% | **54.5%** |
| | 1040M | Baseline | 9.56 | **38.8%** | 80.5% | 89.9% | 37.6% | 64.5% | 38.7% | 58.3% |
| | | MoEUT | **9.36** | 38.0% | **82.5%** | **90.2%** | **38.1%** | **64.6%** | **39.1%** | **58.7%** |

and we found that a minimal regularization is necessary for stabilizing the training. However, larger regularization tends to hurt performance significantly. All other hyperparameters are set based on Tan et al.'s biggest translation experiments. The results are shown in Fig. 7. Effectively, SUTs, which lack our specific methods, have a significant performance disadvantage compared to our MoEUT and the parameter-matched dense baseline. Upon careful investigation, we found that most of this poor performance comes from the ACT mechanism that the authors advertise as one of the main components of their model. After removing the ACT, the performance improves dramatically. However, even with this setup, it underperforms both MoEUT and the standard Transformer baseline. This is also confirmed on downstream tasks in Tab. 2 in the appendix. Moreover, as we show in Appendix A.7, our model runs much faster and uses only a fraction of the memory required for the SUT. To the best of our knowledge, we are not aware of any prior UT architectures that are both competitive and efficient in language modeling.

**Evaluating layer grouping.** We investigate the effect of the layer grouping (Sec. 2.3) on our 244M parameter MoEUT model in Fig. 6. Here, $G$ denotes the number of non-shared layers within the group. $G = 2$ corresponds to the model used in all other analyses. $G = 1$ is a fully shared-layer model, without any grouping, and $G = 18$ corresponds to the baseline fully *non-shared* $\sigma$-MoE model [28]. All hyperparameters are identical among all models, except for the number of MLP experts ($N_E$) and attention experts ($N_A$), which are adjusted to match the parameter count of the dense baseline. In Fig. 6, we observe that $G = 2$ is optimal, and the recurrence in the layer dimension is indeed beneficial. Another interesting question is whether the grouping described in Sec. 2.3 and Fig. 1 is the right way to stack layers. Let us call the two layers in the group A and B. The grouping we discussed so far stacks layers in the form of "ABABAB", e.g., for a 6-layer network. An alternative is to first repeat one of the layers multiple times, followed by the repeated version of the other: "AAABBB". The "AABB" column of Fig. 6 shows this setup for our best $G = 2$ model. It can be seen that the grouping proposed in Sec. 2.3 indeed works significantly better. In fact, the AABB-style stacking is almost as bad as not doing grouping at all.

**Evaluating layernorm schemes.** Here we evaluate our "peri-layernorm" scheme (Sec. 2.4). Fig. 8 shows the results. The proposed layernorm scheme consistently performs the best. The gap is more significant for the small models, while for the bigger ones the gains diminish (for the 719M-parameter

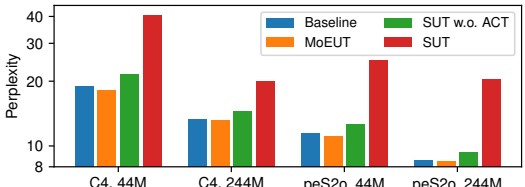
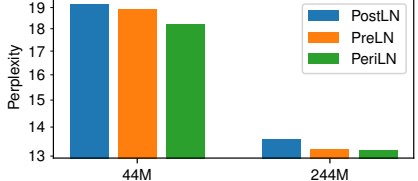

Figure 7: Comparing MoEUT and SUT [38]. MoEUT outperforms both the original SUT and our improved version by a large margin.

Figure 8: Comparing layernorm variants. Peri-layernorm outperforms both pre- and post-layernorm.

model, the gap between peri-norm and post-norm is marginal: 11.29 perplexity points compared to 11.32). At the same time, we also observe that the gap between peri-norm and post-norm increases with the number of training steps, leaving open the possibility of higher gains if the models are trained longer. All our MoEUT models in other experiments make use of this peri-norm scheme.

# 4 Analysis

Here we aim to better understand the learned expert selection of MoEUTs. In what follows, we analyze the expert selection in the MLP blocks (Sec. 2.1) of our 244M parameter MoEUT model trained on C4. All experiments in this section are performed by calculating statistics on the validation set of C4 for a model with $G = 2$ (i.e., two layers in the group; see Sec. 2.3). We only display behaviors of the first layer of the group, as we find that the results of the second layer are qualitatively similar.

**Expert (re)use across layers.** We first focus on whether nontrivial reuse occurs between different layers. Note that *MoE-based* shared-layer models could, in theory, assign different experts to different layers to "emulate" regular non-shared models. If this were the case, the model would resemble a regular Transformer instead of a Universal one. To confirm that our model is more versatile than that, we analyze whether certain experts in the MLP layers are activated only in specific layers. We measure how many times each expert is activated in each layer. This allows us to visualize the distribution of layers that each expert prefers. To better visualize the structure, we reorder experts by a heuristic which we call "layer position score" defined as the average of the layer indices weighted by the number of expert activations in that layer. Fig. 9 shows the results. The yellow spot in the bottom right corner indicates that some experts are assigned mostly to the final layer. However, for the other experts, there is a wide range of layers where the expert is activated. Experts seem to be active in a continuous sequence of layers. This can be seen in the wide, vertically lined structure. We can conclude that MoEUT is capable of specializing in a specific layer if necessary and sharing weights between them when advantageous.

**Per-token expert selection diversity.** Here we analyze the diversity of expert selection in the MLP layers for a given input token across different layers and contexts. For this we measure the total number of unique experts activated for individual tokens at different layers across different positions/contexts. The result is shown in Fig. 10. On the x-axis, the tokens are ordered differently for each layer based on the number of experts used in that layer. We display the most "specialized" 1000 tokens. The minimum possible number of active experts is 16, corresponding to $K$. If only $K$ experts are consistently used by a token across different contexts, it means that the token is fully "specialized" to consistently use a single set of $K$ experts. This is almost the case for many tokens if we look at the first layer (blue curve): the number of unique experts used is low, i.e., the selection mainly depends on the token's identity. However, in the subsequent layers, the situation is quite different: the total number of experts used increases significantly, indicating that the context of the token is taken into account for the expert selection. The diversity of the experts used peaks in the middle layers (here Layer 9) and falls slightly for layers closer to the output. For the converse analysis of expert specialization to tokens/layers, we refer to Appendix A.6.

**Expert selection dynamics in individual columns/positions.** So far, all the results have been cumulative statistics in different input sequences and positions. We might

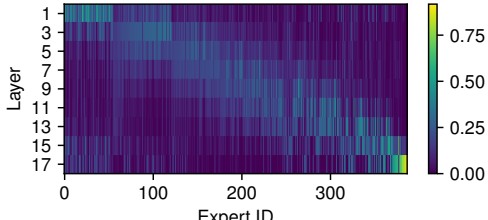

Figure 9: Layer preference of different experts. Most experts are used in multiple layers, while some of them (see, bottom right) specialize to certain layers, showing the flexibility of our model.

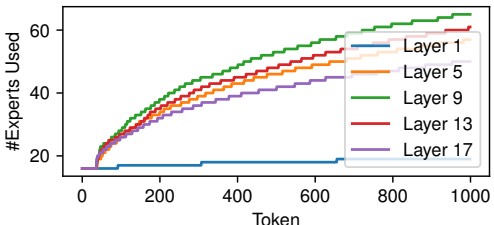

Figure 10: No. of unique experts used in different layers. Tokens are routed to many different experts (depending on the context), especially in the middle layers.

wonder about the selection behavior for the *individual* Transformer columns.[3] Is expert selection mostly constant throughout the layers for individual columns of MoEUT?

To answer this question, we calculate the pairwise intersection-over-union of the set of selected experts between all layers in *individual columns* and average this metric over the whole validation set. We show the result in Fig. 11. There is a non-negligible overlap between the selected experts in subsequent layers; however, it is far from complete overlap. This indicates that experts usually change dynamically in a single column, performing different functionality in different layers.

**Overall**, our analysis suggests that MoEUT is capable of dynamically adapting its expert selection mechanism to a diverse set of circumstances. Sometimes, experts are assigned to popular tokens, while in other cases, they are shared or specialized between layers, depending on what is better for the task.

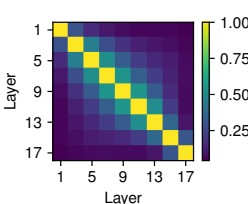

Figure 11: Instance-level average expert selection similarity between layers. Individual tokens are routed to a diverse set of experts across the layers.

## 5 Discussion and Limitations

**More background on UT.** We emphasize that our focus is on developing scalable and performant Universal Transformers for parameter-dominating tasks such as language modeling. This has been a long-standing limitation of UTs. For example, Lan et al. [50] study shared-layer BERT models [51] and find that layer-sharing hurts performance in exchange for better parameter efficiency. Kaplan et al. [19] report that even though Transformer language models with inter-layer parameter-sharing scale better in terms of number of parameters, they fail to achieve compute-efficiency—in contrast, our MoE-based approach is more compute-efficient than the corresponding dense baseline. On the other hand, UTs have been well-known for their compositional generalization capabilities. We refer the readers to the numerous corresponding works (e.g., [16, 15, 38, 52, 14, 10]) for results supporting the benefits of layer sharing. Future work may also use our MoEUT in such compositional settings as a generally more efficient UT architecture.

**MoE for Transformer language models.** MoE methods for Transformer language models have seen many recent advances. It is worth noting that despite many works on MoEs for Transformers (see, e.g., [24, 25, 26, 27, 28]), many of them have only focused on applying MoE to the feedforward layers. Notable exceptions are mixture-of-attention [27] and SwitchHead [31] (Sec. 2.2), which focus on MoE self-attention layers. In addition, until recently, it has been considered challenging to make MoE-based language models competitive against the dense baseline in the parameter-matched setting (unlike FLOPs/MAC-matched settings). In MoEUT, we use $\sigma$-MoE [28], an MoE design that has been shown to be competitive even in such a setting.

---

[3]By representing the Transformer's activations in a 2D grid, with token positions on the x-axis and depth on the y-axis, "columns" correspond to all hidden activations across depth given a token position.

**Further related works on LayerNorm and layer grouping.** There are other works that are closely related to ours regarding certain aspects of our model. Regarding the signal propagation and layernorm [35] in Transformers (Sec. 2.4), Xie et al. [53] analyze the growing residual norm in standard Transformers, and propose a dual, hybrid residual stream as a remedy. Regarding layer grouping, Takase and Kiyono [54] study various layer grouping variants to improve the efficiency of shared-layer Transformers, also showing that layer grouping outperforms vanilla Universal Transformers. However, they consider models with large group sizes ($G = 6$ for 12 layers) and few recurrent steps (2). We find that models with smaller $G$ with more steps perform better. Sometimes, layer grouping is also used to up-scaling pretrained models [55].

**Limitation/Implementation.** Our current implementation of the MoE layers uses the Triton kernel released with $\sigma$-MoE [31] for both the attention and the MLP parts of the model. This implementation is known to be suboptimal [31]. Compared to the standard Transformer with FlashAttention [56], our MoEUT model trains 1.5–2x slower. We estimate that with a more optimal implementation, the training speed should be close to the dense model, while inference should run faster.

**Massive scaling.** Our experiments used a modest training regime. This led to good models and allowed us to make rigorous comparisons, but scaling to massive training regimes for MoEUT remains an important avenue for future research. Such experiments would inevitably require a very large compute cluster, but the costs could also be mitigated somewhat by work optimizing our CUDA kernel.

# 6 Conclusion

We present MoEUT, a novel Mixture-of-Expert-based Universal Transformer (UT) model that addresses the fundamental limitation of the standard UT in terms of parameter–compute efficiency. MoEUT combines the most advanced MoE techniques for Transformers with our novel layer grouping method and layernorm scheme, which are both shown to be crucial for shared-layer models. Our MoEUT allows for training competitive UTs on parameter-dominated tasks such as language modeling, while being significantly less compute intensive than the baselines without layer sharing. We break this long standing limitation of UTs for the first time. Experimentally our model outperforms dense baselines from 44M to 1B parameter scale on C4, SlimPajama, peS2o, and The Stack datasets. Zero-shot experiments confirm that the performance of MoEUT holds on downstream tasks, including BLiMP, CBT, Lambada, HellaSwag, PIQA and ARC-E. We hope that this work helps revive research interest in Universal Transformers at larger scales, and serves as a stepping stone for achieving the superior generalization properties of UTs (typically limited to synthetic problems for now) in real-world settings.

## Acknowledgements

Christopher D. Manning is a CIFAR Fellow. This research was partially funded by ERC Advanced grant no: 742870, project AlgoRNN. We are thankful for hardware donations from NVIDIA and IBM. We are thankful to IDSIA for providing part of the compute used for this project even after the authors left the lab.

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

# A Appendix

## A.1 Broader Impact Statement

We consider this work to be a foundational research paper with no direct societal implications. However, novel research that builds on this work may break the generalization bottleneck of current models, allowing better reasoning. This can potentially be a jump towards Artificial General Intelligence, which might have unforeseeable consequences, both positive and negative. Additionally, a better implementation of our CUDA kernels might lead to foundation models that are more efficient than current ones, which might make them more accessible. This can be beneficial because of the reduction in energy usage, but it might also enable easier generation of harmful content like fake news or deepfakes.

## A.2 Growing Residual Norm In Standard Transformers

In Sec. 2.4, we discussed the issue of the growing residual norm in the standard Transformers. Here, we measure the $L^2$ norm of the difference of the residual before and after applying a standard Transformer layer (both the attention and the MLP block) in different layers of our 44M parameter Transformer trained on C4. The results are visualized in Fig. 12. It can be seen that the norm of the updates indeed grow in later layers.

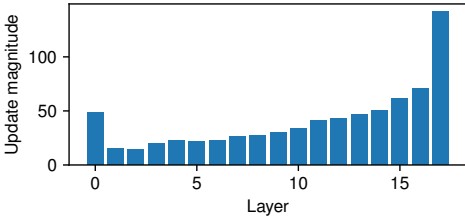

Figure 12: The update magnitude of different layers in a 44M parameter Transformer on C4. The norm of the updates grows throughout the layers to compensate for the residual growth (see Sec. 2.3 for more details).

## A.3 Zero-Shot Downstream Performance of SUT-variants

In addition to evaluating the perplexity of different SUT variants in Fig. 7, we also show their zero-shot downstream performance in Tab. 2. It can be seen that MoEUT consistently outperforms both SUT and SUT without ACT.

Table 2: Zero-shot downstream performance and perplexity of different SUT variants compared to MoEUT and the unshared baseline. MoEUT outperforms both SUT variants.

| Dataset | #params | Model | PPL ↓ | LAMBADA ↑ | BLiMP ↑ | CBT ↑ | HellaSwag ↑ | PIQA ↑ | ARC-E ↑ | Average ↑ |
|---|---|---|---|---|---|---|---|---|---|---|
| C4 | 44M | Baseline | 18.97 | 21.9% | 73.5% | 81.3% | 28.3% | 59.9% | 31.7% | 49.4% |
| | | MoEUT | **18.30** | **23.2%** | **78.2%** | **81.1%** | **29.2%** | **61.3%** | **33.5%** | **51.1%** |
| | | SUT | 40.50 | 1.2% | 65.3% | 51.1% | 26.4% | 57.8% | 31.9% | 39.0% |
| | | SUT w.o. ACT | 21.51 | 18.1% | 72.8% | 66.3% | 27.5% | 59.1% | 32.5% | 46.0% |
| | 244M | Baseline | 13.40 | **33.1%** | 78.5% | **86.0%** | 34.5% | 64.9% | **36.9%** | **55.6%** |
| | | MoEUT | **13.24** | 30.6% | **79.7%** | 85.3% | **35.7%** | **65.2%** | 36.4% | 55.5% |
| | | SUT | 20.05 | 20.5% | 71.0% | 68.5% | 28.2% | 60.1% | 32.7% | 46.8% |
| | | SUT w.o. ACT | 14.58 | 27.8% | 77.0% | 75.9% | 32.7% | 63.2% | 35.5% | 52.0% |
| peS2o | 44M | Baseline | 11.46 | **13.2%** | 66.5% | 68.6% | **28.5%** | 56.3% | **32.0%** | 44.2% |
| | | MoEUT | **11.09** | 13.1% | **68.7%** | **69.6%** | 28.3% | 55.1% | 31.4% | **44.4%** |
| | | SUT | 25.04 | 0.5% | 59.2% | 38.1% | 26.2% | 55.0% | 31.1% | 35.0% |
| | | SUT w.o. ACT | 12.68 | 11.7% | 66.5% | 53.9% | 28.0% | 56.1% | 31.5% | 41.3% |
| | 244M | Baseline | 8.55 | 18.7% | 72.8% | **78.0%** | **30.4%** | 56.3% | 35.0% | 48.5% |
| | | MoEUT | **8.52** | **19.4%** | **73.5%** | 77.4% | 30.1% | 56.3% | **35.6%** | **48.7%** |
| | | SUT | 20.44 | 0.5% | 60.9% | 42.8% | 26.7% | 55.3% | 33.0% | 36.5% |
| | | SUT w.o. ACT | 9.31 | 16.8% | 71.9% | 64.8% | 28.8% | **57.3%** | 34.8% | 45.7% |

**A.4 Hyperparameters**

All our models are trained in PyTorch [57] with a batch size of 64, context length of 1024, for 100k iterations, a learning rate of 0.00025, AdamW optimizer [58] with default hyperparameters, weight decay of 0.01. They are trained on a single node in a data-parallel manner. The learning rate is decayed to $10\%$ of its initial value using cosine decay. We use a gradient clipping of $\kappa$ and $N_{\text{warmup}}$ linear learning rate warmup steps (see Tab. 3). None of our models uses dropout. For the entropy regularization of the MLP expert selection, we use $\gamma = 0.01$ and for SwitchHead attention $\delta = 0.001$. Expert dropout is not used. All of our models use a SentencePiece [59] tokenizer with 8000 tokens, trained on a subset of the training set for the given dataset. All models are trained with mixed precision. The hyperparameters of the SUT models can be found in Tab. 4. Note that the meanings of the parameters are not directly analogous to ours. Please refer to Tan et al. [38] for more details.

Table 3: Hyperparameters of different models used in our main experiments.

| Model | #params | $n_{\text{layers}}$ | $G$ | $d_{\text{model}}$ | $d_{\text{ff}}$ | $H$ | $N_A$ | $d_{\text{head}}$ | $N_E$ | $K$ | $N_{\text{warmup}}$ | $\kappa$ |
|---|---|---|---|---|---|---|---|---|---|---|---|---|
| Baseline | 45M | 16 | - | 412 | 2053 | 10 | - | 41 | - | - | 0 | 0.1 |
| MoEUT | 44M | 16 | 2 | 412 | - | 4 | 8 | 82 | 155 | 12 | 0 | 0.1 |
| $\sigma$-MoE | 44M | 16 | 16 | 412 | - | 4 | 1 | 82 | 17 | 12 | 0 | 0.1 |
| Baseline | 126M | 16 | - | 768 | 3072 | 16 | - | 48 | - | - | 4000 | 0.25 |
| MoEUT | 126M | 18 | 2 | 768 | - | 4 | 10 | 96 | 254 | 12 | 4000 | 0.25 |
| $\sigma$-MoE | 126M | 18 | 18 | 768 | - | 4 | 1 | 96 | 26 | 12 | 4000 | 0.25 |
| Baseline | 244M | 18 | - | 1024 | 4110 | 16 | - | 64 | - | - | 4000 | 0.25 |
| MoEUT | 243M | 18 | 2 | 1024 | - | 4 | 10 | 128 | 387 | 16 | 4000 | 0.25 |
| $\sigma$-MoE | 244M | 18 | 18 | 1024 | - | 4 | 1 | 128 | 40 | 16 | 4000 | 0.25 |
| Baseline | 319M | 24 | - | 1024 | 4110 | 16 | - | 64 | - | - | 4000 | 0.25 |
| MoEUT | 318M | 24 | 3 | 1024 | - | 4 | 10 | 128 | 338 | 16 | 4000 | 0.25 |
| $\sigma$-MoE | 320M | 24 | 24 | 1024 | - | 4 | 1 | 128 | 40 | 16 | 4000 | 0.25 |
| Baseline | 729M | 36 | - | 1280 | 5120 | 20 | - | 64 | - | - | 4000 | 0.25 |
| MoEUT | 727M | 36 | 4 | 1280 | - | 5 | 13 | 128 | 467 | 20 | 4000 | 0.25 |
| $\sigma$-MoE | 731M | 36 | 36 | 1280 | - | 5 | 1 | 128 | 50 | 20 | 4000 | 0.25 |
| Baseline | 1044M | 36 | - | 1536 | 6144 | 24 | - | 64 | - | - | 4000 | 0.25 |
| MoEUT | 1040M | 36 | 4 | 1536 | - | 6 | 12 | 128 | 565 | 24 | 4000 | 0.25 |

Table 4: Hyperparameters of SUT models used in our experiments.

| #params | $n_{\text{layers}}$ | $d_{\text{model}}$ | $d_{\text{expert}}$ | $H$ | $N_A$ | $d_{\text{att\_expert}}$ | $d_{\text{head}}$ | $N_E$ | $K$ | $\mathcal{L}_{\text{MIM}}$ | $\mathcal{L}_{\text{ACT}}$ | $N_{\text{warmup}}$ | $\kappa$ |
|---|---|---|---|---|---|---|---|---|---|---|---|---|---|
| 45M | 16 | 412 | 256 | 4 | 24 | 256 | 64 | 152 | 2 | 0.001 | 0.01 | 0 | 0.1 |
| 245M | 18 | 1024 | 512 | 4 | 21 | 512 | 128 | 192 | 4 | 0.01 | 0.01 | 4000 | 0.25 |

**A.5 The Effects of $d_{\text{expert}}$ and $K$**

Here, we analyze the effect of the expert size given a fixed amount of compute ($d_{\text{expert}} \cdot K$ being kept constant). The results are shown in Fig. 13. It can be seen that using fine-grained mixture of experts (small experts) is indeed critical for good performance. In our models, we use $d_{\text{expert}} = 128$. Using smaller experts significantly decreases the compute efficiency of the Tirton kernel. Please note that these experiments keep the number of active channels in the MLP block constant. Thus, the effect is based purely on the dynamics of the selection mechanism.

We also analyzed the performance of our MoEUT model with different numbers of active experts in the MLP layer. This varies the amount of compute spent in the layer, and the number of active channels. We show the results in Fig. 14. Increasing the number of active experts always improves performance, but the returns diminish. We chose $G = 16$ for our experiments because of efficiency reasons.

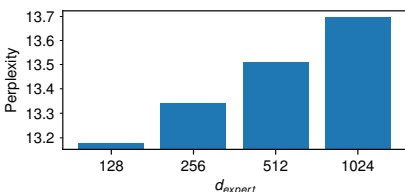

Figure 13: Performance of our 244M MoEUT on a held-out subset of C4 with expert sizes ($d_{\text{expert}}$) in the MLP layer. The smallest expert size performs the best.

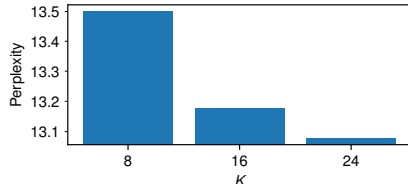

Figure 14: Performance of our 244M MoEUT on a held-out subset of C4 with different number of active experts for the $\sigma$-MoE layer. Increasing the number of experts always helps, but the returns diminish.

## A.6 Additional Analysis

**Expert specialization to tokens/layers.** Now conversely to the "per-token expert selection diversity" analysis presented in the main text, we analyze whether the experts activated by a token are layer-specific for that specific token. For this, we count the number of unique experts used by each token and compute the corresponding proportion for each layer. The results are shown in Fig. 15. Here, we order the tokens (x-axis) by their frequency in the validation set (the same ordering is used for all layers). The first 6000 tokens are shown.

We observe that for the most frequent tokens (toward the left part of the plot), high scores near 1.0 are obtained in multiple layers for a given token; this means that (almost) all experts used by that specific token are used in multiple layers. In contrast, we observe that experts tend to be more layer-specific for the less popular tokens (toward the right part of the plot). In addition, the set of experts selected in the early layers is typically less diverse than for the rest of the layers: only a small fraction of the used experts are present there. This is consistent with the findings for Layer 1 in Fig. 10.

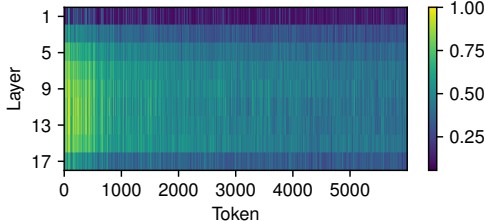

Figure 15: Proportion of experts used in a specific layer out of all unique experts used by a given token. On the x-axis, the tokens are ordered by decreasing frequency of occurrence. The less frequent a tokens is, the more layer-specific are the experts used by that token.

## A.7 Wall Clock Time and Memory Comparison

To show the real-world resource usage of our models directly, we run a controlled experiment on identical hardware with our 244M parameter model and the corresponding baselines.

We measured the training iteration time and memory usage on 8 V100 32GB GPUs. Here one "iteration" corresponds to an effective batch size of 64x1024 tokens for all models. The training

iteration time was measured by using a batch size for each model that fits GPUs; models require either 1 or 2 gradient accumulation steps to achieve the effective batch size depending on their memory requirement. We measured the training time right after initialization and a warmup period. The memory usage is measured using 2 grad accumulation steps for all models for a fair comparison. Note that around 3GB of memory is used by the model parameters and optimizer state on each GPU. We show the results in Tab. 5.

Even though our MoEUT with the current kernel implementation is slower than the corresponding dense non-shared layer transformer, it is significantly faster and uses much less memory compared to alternative UT variants.

Table 5: Wall-Clock Time for the forward-backward pass and the total memory usage of our training loop with different 244M parameter models on 8 V100 GPUs, which a batch size of 64x1024 tokens. MoEUT is 1.7x slower with our suboptimal MoE kernel implementation than the standard transformer, but it is much faster than the other UT variants. It also uses much less memory, allowing training on larger scales.

| Model | ms/batch | Memory usage |
|---|---|---|
| Non-shared Transformer | 443 | 9.2 G |
| Naive UT | 3559 | 25.9 G |
| MoEUT | 772 | 9.0 G |
| SUT | 1344 | 23.4 G |

## A.8 Compute Requirements

We report the hardware used and the required wall clock time for our main experiments in the paper in Tab. 6. All experiments are performed in private clusters. The duration is reported in "hh:mm" format. We report the number of GPUs ($N_{GPU}$) used for that specific experiment (and *not* the total number of GPUs in the system). For the number of CPUs ($N_{CPU}$) and RAM we report the total amount in the node, as these resources are shared between concurrent runs.

Note that the report is generated from Weights and Biases logs. Because of this, the duration might include effects of restarts because of SLURM preemption and effects for occasional slowdowns of the network drive. Additionally, in a few instances the Weights and Biases log buffer overflowed, making it impossible to determine the number of GPUs used for the experiment. In this case, we report "??" for the corresponding experiment. Note that we only report the resource usage of the final experiments here. We estimate that the total cost of the failed experiments and preliminary runs is at least an order of magnitude higher than this.

Table 6: Training hardware information for the experiments reported in the paper

| Model | #params | Dataset | $G$ | GPU Type | $N_{\text{GPU}}$ | $N_{\text{CPU}}$ | RAM | Duration |
|---|---|---|---|---|---|---|---|---|
| SUT | 45M | C4 | - | TITAN RTX | 4 | 8 | 125G | 29:28 |
| SUT | 45M | peS2o | - | RTX 3090 | 5 | 24 | 188G | 28:46 |
| SUT | 245M | C4 | - | A100-80GB | 4 | 128 | 1007G | 28:20 |
| SUT | 245M | peS2o | - | A100-80GB | 4 | 128 | 1007G | 23:43 |
| MoEUT PostLN | 44M | C4 | 2 | RTX 3090 | 5 | 24 | 251G | 13:42 |
| MoEUT PostLN | 243M | C4 | 2 | V100-32GB | 8 | 40 | 503G | 23:40 |
| $\sigma$-MoE | 44M | C4 | - | V100-16GB | ?? | 32 | 220G | 20:09 |
| MoEUT | 44M | C4 | 2 | V100-16GB | 4 | 40 | 251G | 19:32 |
| MoEUT | 44M | peS2o | 2 | V100-16GB | ?? | 32 | 220G | 22:04 |
| MoEUT | 44M | SlimPajama | 2 | RTX 3090 | 5 | 24 | 188G | 15:21 |
| MoEUT | 126M | C4 | 2 | RTX 3090 | 5 | 24 | 251G | 20:05 |
| $\sigma$-MoE | 126M | C4 | - | RTX 3090 | 5 | 24 | 251G | 17:27 |
| MoEUT AABB | 243M | C4 | 2 | RTX 3090 | 5 | 24 | 188G | 41:02 |
| MoEUT | 243M | C4 | 2 | RTX 4090 | 5 | 24 | 251G | 26:03 |
| MoEUT PreLN | 243M | C4 | 2 | V100-32GB-LS | ?? | 40 | 503G | 46:09 |
| MoEUT | 243M | peS2o | 2 | RTX 4090 | 5 | 24 | 251G | 24:59 |
| MoEUT | 243M | SlimPajama | 2 | RTX 3090 | 5 | 24 | 251G | 32:55 |
| MoEUT | 243M | TheStack | 2 | RTX 4090 | 5 | 24 | 251G | 24:36 |
| MoEUT | 244M | C4 | 1 | RTX 3090 | 5 | 24 | 251G | 36:39 |
| MoEUT | 244M | C4 | 3 | RTX A6000 | 5 | 48 | 503G | 11:08 |
| MoEUT | 244M | C4 | 6 | RTX 3090 | 5 | 24 | 251G | 30:06 |
| MoEUT | 244M | C4 | 9 | RTX 4090 | 5 | 24 | 251G | 22:30 |
| $\sigma$-MoE | 244M | C4 | - | V100-32GB-LS | 8 | 40 | 503G | 19:34 |
| MoEUT | 318M | C4 | 3 | V100-32GB | 8 | 40 | 503G | 27:51 |
| $\sigma$-MoE | 320M | C4 | - | RTX 3090 | 5 | 24 | 251G | 37:06 |
| MoEUT | 727M | C4 | 4 | A100-80GB | 4 | 128 | 1007G | 53:28 |
| MoEUT | 727M | TheStack | 4 | A100-80GB | 4 | 128 | 1007G | 38:40 |
| $\sigma$-MoE | 731M | C4 | - | V100-32GB-LS | 8 | 40 | 503G | 52:27 |
| MoEUT | 1040M | C4 | 4 | A100-80GB | 4 | 128 | 1007G | 74:10 |
| MoEUT | 1040M | SlimPajama | 4 | A100-80GB | 4 | 128 | 1007G | 70:35 |
| Transformer | 45M | C4 | - | RTX 3090 | 2 | 48 | 251G | 24:08 |
| Transformer | 45M | peS2o | - | V100-16GB | 4 | 32 | 220G | 13:32 |
| Transformer | 45M | SlimPajama | - | V100-16GB | 4 | 40 | 251G | 11:59 |
| Transformer | 126M | C4 | - | RTX A6000 | 4 | 48 | 503G | 12:03 |
| Transformer | 244M | C4 | - | V100-32GB | 8 | 40 | 503G | 14:22 |
| Transformer | 244M | peS2o | - | RTX A6000 | 4 | 48 | 503G | 21:04 |
| Transformer | 244M | SlimPajama | - | V100-32GB-LS | 8 | 40 | 503G | 19:08 |
| Transformer | 244M | TheStack | - | RTX 3090 | ?? | 24 | 251G | 26:15 |
| Transformer | 319M | C4 | - | V100-32GB | 8 | 40 | 503G | 16:36 |
| Transformer | 729M | C4 | - | V100-32GB | 8 | 40 | 503G | 31:29 |
| Transformer | 729M | TheStack | - | A100-80GB | 4 | 128 | 1007G | 25:39 |
| Transformer | 1044M | C4 | - | A100-80GB | 4 | 128 | 1007G | 31:26 |
| Transformer | 1044M | SlimPajama | - | A100-80GB | 4 | 128 | 1007G | 31:38 |

