# OpenReview forum: "MoEUT: Mixture-of-Experts Universal Transformers"
_NeurIPS.cc/2024/Conference — NeurIPS 2024 poster_

### Official Review · Reviewer_gEb4 · 2024-07-07

**Soundness:** 3
**Presentation:** 3
**Contribution:** 3
**Rating:** 6
**Confidence:** 4

**Summary:**

Motivated by the superior generalization performance of Universal Transformers (UT) demonstrated in other works, this paper addresses the compute efficiency problem of this architecture. While UT decreases the parameter count drastically by sharing parameters across layers, vanilla UT underperforms dense transformers on typical NLP tasks. It is unclear how to efficiently scale the parameter count of the UT, as scaling a single layer to compensate for the loss of parameters would result in computational costs beyond those of standard transformers due to increased layer width.

First, this paper explores Mixture-of-Experts (MoE) techniques to efficiently scale the parameters of the UT. Due to sparse expert activation, MoE can scale up the parameter count without drastically increasing the compute requirements. Since simply scaling the parameter count with MoE layers does not yield the expected performance gains, the authors propose two additional innovations: Layer grouping—where parameters are shared within groups of layers instead of across all layers—and Peri-layernorm—where layer normalization is applied only before linear projections that are followed by sigmoid or softmax activation functions.

At a high level, my understanding is that the proposed MoEUT architecture essentially interpolates between a dense transformer without shared layers and a UT with completely shared layers. MoEUT can learn to reuse the same experts for each layer (or group), which corresponds to vanilla UT, or it can learn to use completely different, non-overlapping experts at each layer, resembling a standard transformer.

**Strengths:**

Overall it is an interesting paper, the proposed method is (conceptually) simple, sound and well motivated. I especially like that authors also included zero-shot downstream task evaluation in the empirical section.

Originality: the paper builds on existing knowledge in a meaningful way. It uses existing MoE technique to efficiently increase capacity of UT, which is not a new idea as it had already been proposed in the SUT paper (as aknowledged and discussed by the authors). It adds two additional ideas that seem to have a positive effect on the performance: layer grouping and "peri-layernorm". Yet, as discussed in chapter 5, also layer-grouping has been investigated in prior works.

Quality & Clarity: overall, the claims of the paper are supported by evidences. The paper is adequately written, though chapters 2.1 and 2.2 might be dense for readers unfamiliar with the two architectures the authors build upon. The arguments are mostly clear with some exceptions (see Questions).

Significance: good. the paper addresses relevant problem and makes a moderate contribution to the field. This work has the potential to influence subsequent studies on scaling UT architectures.

**Weaknesses:**

At a high level, my main concern is the following: given that MoEUT effectively interpolates between a standard transformer and vanilla UT, do the authors think that MoEUT would still keep the advantage of vanilla UT when it comes to systematic generalization?
Beyond that, I have some additional questions/doubts that I list in the questions part.

**Questions:**

- ll. 28 - 29: the fact that UT shines in systematic generalization baselines does not make it a more "general" architecture, does it? In contrary, UT imposes the requirement of parameter sharing across layers, which can be see as an additional inductive bias, which could be potentially learned by the standard transformer from the data.
- Chapter 3: I am a bit confused by the fact that σ-MoE underperforms the dense transformer baseline in parameter-matched setting. (Fig.4a), which seem to be in contradiction with the results from the σ-MoE paper. Why is it the case?
- l. 172: my understanding from the related literature is that relative positional encoding is essential to enable systematic generalization in UT. Why do authors decide to use RoPE positional encodings here?
- l. 219: "We note that SUTs have not been evaluated previously on any language modeling tasks." -- the original SUT was evaluated on e.g. English-German translation task, Is it not a language modelling task?
- ll. 319 - 322: Authors talk about importance of comparing parameter-matched MoE and dense models. IT would be useful if they could explain why parameter-matched setting is important (as compared to compute matched)
- ll. 287 - 299: Does "column" refer to individual token representations throughout the layers in transformer?

**Limitations:**

Authors adequately address the limitations in chapter 5.

---

> ### Author Rebuttal · Authors · 2024-08-05
>
> We would like to thank the reviewer for the valuable review and for positive comments on the methodology of the paper. Please find our responses as follows:
>
> > .. given that MoEUT effectively interpolates between a standard transformer and vanilla UT, do the authors think that MoEUT would still keep the advantage of vanilla UT when it comes to systematic generalization?
>
> We can confirm that: we evaluated MoEUT with G=2 on the publicly available dataset of [1] for learning random knowledge graphs. Similarly to Fig. 14 in the appendix of [1], our model learns ~70% OOD generalization. In contrast, the standard Transformers completely fail.
>
> > the fact that UT shines in systematic generalization baselines does not make it a more "general" architecture, does it? In contrary, UT imposes the requirement of parameter sharing across layers, which can be see as an additional inductive bias, which could be potentially learned by the standard transformer from the data.
>
> We would first like to clarify that UTs are in principle strictly more general than *parameter equivalent* standard transformers: they could assign Nparams/Nlayers of their weights to each layer deterministically and simulate a standard Transformer. In this sense, UTs are not at all a “restricted version” of the standard Transformers.
>
> That said, the reviewer’s remark regarding the standard Transformer and learning layer-sharing from the data is a valid point. In practice with finite amounts of data, however, learning such behavior purely from data is very inefficient, if not impossible. A good illustration of this is compositional tasks. Transformers typically learn to solve compositional tasks by allocating each step/function to each layer. In realistic settings, not all compositional combinations are present in the dataset, thus, certain functions are only learned in certain layers; causing failures to generalize on unseen compositions.
>
> In fact, [1] rigorously analyzes this in synthetic knowledge graphs, unveiling that transformers use early layers for resolving the first hop, and later layers to resolve the second. However, the knowledge about rarely-seen compositions will be only available in the early layers, thus they can’t be composed with others. [2] shows similar issues on LLama3-70b on real world problems.
>
> > I am a bit confused by the fact that σ-MoE underperforms the dense transformer baseline in parameter-matched setting. (Fig.4a), which seem to be in contradiction with the results from the σ-MoE paper. Why is it the case?
>
> Please note that our baselines are much stronger than those reported in the 𝜎-MoE paper: on C4, we achieve a perplexity of 13.4 using 244M parameters vs. 17.79 reported by the 𝜎-MoE paper using 266M.
>
> This difference comes from two modifications. First, the 𝜎-MoE paper follows the experimental protocol of Transformer XL: we used their official 𝜎-MoE codebase, but improved their baseline by using RoPE and no XL cache. Second, they use dropout in the FFN layers of their baseline, and ‘expert dropout’ in the 𝜎-MoE. Here we disabled all dropouts in all our models as we use sub-epoch training. This resulted in perplexity improvements with a higher gain for the baseline than for 𝜎-MoE.
>
> > Why do authors decide to use RoPE positional encodings here?
>
> RoPE is a form of relative positional encoding and is used by most modern LLMs, like LLama. Given its popularity, we considered it the best choice.
>
>
> > "We note that SUTs have not been evaluated previously on any language modeling tasks." -- the original SUT was evaluated on e.g. English-German translation task, Is it not a language modelling task?
>
> We can soften or remove this claim, since it depends a lot on how one approaches the problem. Translation tasks (like many NLP tasks) can be formulated as an LM task, but in practice, the translation benchmarks (used in the original SUT paper) and those for evaluating LLMs (used in ours) are very different in important ways. The former is an isolated task where people explicitly train models on a dedicated translation dataset (even a specific encoder-decoder architecture is often used instead of generic decoder-only auto-regressive models). This makes translation benchmarks somewhat easier in the sense that very small, specialized models perform pretty well (e.g., the Transformer baseline used in the SUT paper only has 65M parameters; while their biggest SUT also only has 110M params).
>
> > IT would be useful if they could explain why parameter-matched setting is important (as compared to compute matched)
>
> The parameter-matched setting is crucial to evaluate the model’s *expressiveness* in the LLM tasks where the number of parameters has a high impact on the model performance. We consider this setting to be particularly important to evaluate the true expressiveness of MoEs compared to their dense counterparts.
>
> While the compute-matched setup has values when considering certain practical settings, it gives an “unfair” advantage to MoEs in terms of comparison, as we can easily add extra parameters to an MoE without significantly increasing compute requirements. Here we wanted to show that our MoEUT is capable, even without considering such an advantage, by evaluating its pure expressiveness in the more challenging parameter-matched setting.
>
> > Does "column" refer to individual token representations throughout the layers in transformer?
>
> Yes, it does. We thank the reviewer for pointing out this ambiguity and will improve the clarity in the final version.
>
> We believe our response above resolves all the concerns that the reviewer has raised. If the reviewer finds our response useful, please consider increasing the score. Thank you very much.
>
> [1] Wang et al: Grokked Transformers are Implicit Reasoners: A Mechanistic Journey to the Edge of Generalization
>
> [2] Biran et al: Hopping Too Late: Exploring the Limitations of Large Language Models on Multi-Hop Queries

---

> > ### Comment · Reviewer_gEb4 · 2024-08-09
> >
> > I would like to thank the authors for their comprehensive replies. I appreciate that the authors tested MoEUT on systematic generalization tasks (learning random graphs), making the method even more convincing.

---

> > > ### Author Response · Authors · 2024-08-09
> > >
> > > Thank you very much for your response! We are glad to hear that the reviewer found our response useful!

---

### Official Review · Reviewer_ZbxN · 2024-07-10

**Soundness:** 2
**Presentation:** 3
**Contribution:** 2
**Rating:** 6
**Confidence:** 4

**Summary:**

This paper introduces a novel application of the mixture of experts (MoE) architecture within both the MLP and attention modules of the Universal Transformer network. The integration of MoE is further complemented by an innovative sequence-level routing regularization technique, which the authors argue enhances training stability. Additionally, the paper proposes the incorporation of layer grouping and a new layer normalization scheme, aimed at boosting model performance. Compared with baselines, the proposed MoEUT can achieve better perplexity with the same number of parameters and obtain competitive performance in various downstream tasks.

**Strengths:**

1. The paper demonstrates the efficacy of incorporating the Mixture of Experts (MoE) architecture into a shared-layer Transformer network, highlighting its potential to improve performance.
2. By integrating the MoE model with layer grouping and a novel layer normalization approach, the proposed model achieves superior results in language modeling tasks compared to standard Transformers.
3. Visualization of the results reveals significant specialization across different layers, with each layer showing a distinct preference for particular experts, indicating effective learning and specialization within the network.

**Weaknesses:**

1. The relationship between the MoE network architecture, the proposed layer grouping strategy, and the novel layer normalization technique appear to be undefined. Clarification on how these components synergize within the model would enhance understanding of their collective impact on performance.

2. While the author suggests that the $\sigma$-MoE model exhibits instability during training, this assertion is not substantiated with empirical evidence. Providing experimental results or a more detailed analysis to support this claim would strengthen the argument.

3. The current ablation study does not sufficiently demonstrate the effectiveness of the proposed method. Expanding the ablation study to include a broader range of experiments and comparative analyses could offer a more comprehensive evaluation of the individual contributions of the proposed enhancements.

**Questions:**

1. **Layer Grouping and Novel Layer Norm Performance Contribution:**
   - Do the layer grouping and the novel layer normalization contribute independently to the performance improvements observed, or are these contributions specifically related to the architecture of the Mixture of Experts (MoE) network?

2. **Impact of Improved Sequence Level Routing Regularization on Training Stability:**
   - Does the implementation of improved sequence-level routing regularization enhance the training stability of the model?

3. **Distinct Contributions of MoE Variants in MoEUT:**
   - Given the implementation of MoE within the MoEUT framework (specifically in the MLP and attention mechanisms), how does each variant individually affect the model's overall performance?

4. **Impact of Excluding MoE on Performance:**
   - If the MoE design is omitted, allowing layer grouping and “peri-layernorm” to independently influence the model, what is the anticipated impact on performance? Additionally, is the integration of layer grouping and “peri-layernorm” with the proposed MoE architecture necessary for achieving the observed benefits?

5. **Comparative Analysis of MoEUT and SUT:**
   - In addition to perplexity comparisons, it is suggested that a direct comparison between MoEUT and SUT (Sparse Transformer) should also be conducted across the downstream language modeling tasks presented in Table 1. This would provide a more comprehensive understanding of their relative performances.

**Limitations:**

The authors adequately discuss the limitations and do not have the potential negative societal impact.

---

> ### Author Rebuttal · Authors · 2024-08-05
>
> We would like to thank the reviewer for their valuable time reviewing our work. We would like to respond to the concerns raised by the reviewer as follows.
>
>
> > … . Clarification on how these components synergize within the model would enhance understanding of their collective impact on performance…
>
> > Do the layer grouping and the novel layer normalization contribute independently to the performance improvements observed, or are these contributions specifically related to the architecture of the Mixture of Experts (MoE) network?
>
> > If the MoE design is omitted, allowing layer grouping and “peri-layernorm” to independently influence the model, what is the anticipated impact on performance? Additionally, is the integration of layer grouping and “peri-layernorm” with the proposed MoE architecture necessary for achieving the observed benefits?
>
> We would first like to draw the reviewer’s attention to the fact that we have dedicated ablation studies on the effect of layer grouping in Fig. 6 and the effect of different layer normalizations in Fig. 8 independently of each other for our MoE-based model. We demonstrated that both help.
>
>
> Testing these on the naive UTs without MoE in a systematic/fair way is not easy precisely because of the challenge of scaling the naive UTs (the very problem we address in this work): they are prohibitively slow and use too much memory for any interesting scale. Given that MoE is anyway necessary to scale up UTs, it seemed natural to focus on evaluating these methods for the MoE-based UTs.
>
> Also intuitively, we have no reason to believe that the contributions of these methods would be very different for UTs because the original motivations of these methods are based on UT’s properties, and they are unrelated to the use of MoE (e.g., peri-layernorm is motivated by the residual norm growth caused by parameter-sharing across layers; see A.2).
>
>
> > The current ablation study does not sufficiently demonstrate the effectiveness of the proposed method. Expanding the ablation study to include a broader range of experiments and comparative analyses could offer a more comprehensive evaluation of the individual contributions of the proposed enhancements.
>
>
> We believe that the ablation studies presented in the current paper sufficiently cover the most important aspects to justify our design choice within MoEUT. Please see Fig 6 for the effect of routing, Fig 8 for comparing pre/post/peri layernorm, Fig 13 for the effect of d_expert, and Fig 14 for the effect of K.
> As we explained above, conducting further ablations on the naive UTs without MoE is not reasonable because of their scale inefficiency. If the reviewer still thinks there are any other ablations that are critically missing, we would appreciate it a lot if the reviewer could suggest concrete ideas. Thank you.
>
> > While the author suggests that the 𝜎-MoE model exhibits instability during training, this assertion is not substantiated with empirical evidence. Providing experimental results or a more detailed analysis to support this claim would strengthen the argument.  … Does the implementation of improved sequence-level routing regularization enhance the training stability of the model?
>
> Yes, without sequence-level routing regularization, larger models suffer from an expert collapse and they diverge. We found it uninformative to include training loss curves exploding to infinity in the appendix, but if the reviewer thinks that would be interesting for the readers, we can include them in the appendix of the next version of the paper.
>
> > In addition to perplexity comparisons, it is suggested that a direct comparison between MoEUT and SUT (Sparse Transformer) should also be conducted across the downstream language modeling tasks presented in Table 1. This would provide a more comprehensive understanding of their relative performances.
>
>
> Thank you for pointing this out. We extended our downstream task evaluations to SUT and “SUT without ACT” (in the meanwhile, we identified that the “ACT” component of SUT, which is a fundamental component of SUT, is detrimental to its performance here). These results show that MoEUT achieves 18-33% increases (8.7-12.2 points) over SUT. Removing ACT from SUT improves its downstream performance as well, but MoEUT remains consistently better, with 7-11% increases (3-5.1 points) on average across our tasks, model sizes, and pretraining datasets.
>
> | Dataset | #params | Model |  PPL | LAMBADA | BLiMP | CBT | HellaSwag | PIQA | ARC-E | Average |
> |----|----|----|----|----|----|----|----|----|----|----|
> | C4 |  44M | MoEUT | 18.30 | 23.2% | 78.2% | 81.1% | 29.2% | 61.3% | 33.5% | 51.1% |
> | | 44M | SUT | 40.50 | 1.2% | 65.3% | 51.1% | 26.4% | 57.8% | 31.9% | 39.0% |
> | | 44M | SUT w.o. ACT | 21.51 | 18.1% | 72.8% | 66.3% | 27.5% | 59.1% | 32.5% | 46.0% |
> | | 244M | MoEUT | 13.24 | 30.6% | 79.7% | 85.3% | 35.7% | 65.2% | 36.4% | 55.5% |
> | | 244M | SUT | 20.05 | 20.5% | 71.0% | 68.5% | 28.2% | 60.1% | 32.7% | 46.8% |
> | | 244M | SUT w.o. ACT | 14.58 | 27.8% | 77.0% | 75.9% | 32.7% | 63.2% | 35.5% | 52.0% |
> | PES2O | 44M | MoEUT | 11.09 | 13.1% | 68.7% | 69.6% | 28.3% | 55.1% | 31.4% | 44.4% |
> | | 44M | SUT | 25.04 | 0.5% | 59.2% | 38.1% | 26.2% | 55.0% | 31.1% | 35.0% |
> | | 44M | SUT w.o. ACT | 12.68 | 11.7% | 66.5% | 53.9% | 28.0% | 56.1% | 31.5% | 41.3% |
> | | 244M | MoEUT | 8.52 | 19.4% | 73.5% | 77.4% | 30.1% | 56.3% | 35.6% | 48.7% |
> | | 244M | SUT | 20.44 | 0.5% | 60.9% | 42.8% | 26.7% | 55.3% | 33.0% | 36.5% |
> | | 244M | SUT w.o. ACT | 9.31 | 16.8% | 71.9% | 64.8% | 28.8% | 57.3% | 34.8% | 45.7% |
>
> We will add this to the updated version of our paper.
>
> We believe our response above resolves all the concerns that the reviewer has raised. If the reviewer finds our response useful, please consider increasing the score. Thank you very much.

---

> > ### Comment · Reviewer_ZbxN · 2024-08-12
> > **Official Comment by Reviewer ZbxN**
> >
> > I extend my gratitude to the authors for their detailed responses. To fully grasp the efficacy and interplay of the proposed components, I anticipate an exploration of the performance for MoEUT without PeriLN and vice versa. Such results would provide a clearer understanding of these elements.
> >
> > Moreover, I am keen to understand how MoEUT and PeriLN complement each other, suggesting that their integration is not merely coincidental but inherently synergistic.
> >
> > The authors have addressed my concerns to some extent, therefore I would like to increase my score. Should the authors further elaborate on these aspects, specifically by demonstrating the independent and combined effects of MoEUT and PeriLN, I am inclined to further improve their score.

---

> > > ### Author Response · Authors · 2024-08-13
> > >
> > > We are very thankful for the increased score! We are glad to hear that the reviewer found our response useful!
> > >
> > > We agree with the reviewer that analyzing PeriLN in more detail would be a very useful additional experiment for our paper. First, we would like to emphasize that one direction of such comparison is already part of the paper: we analyze the effect of different layer norm variants on MoEUT in Figure 8 of the paper. The reverse direction is missing: We do not have experiments with PeriLN on standard (naive) UTs. We tried to run such experiments upon the suggestion of the reviewer. We would like to emphasize that this is very resource intensive: we need at least 8 A6000-level GPUs to make the experiment fit in the memory even for our tiny, 44M naive UT model. Unfortunately, we found that naive UTs experience residual blowups and become unstable during training.
> > > Such blowup does not happen in MoEs, possibly because the number of simultaneously active channels/attention heads is limited to a much smaller number than in naive UTs, where all of them can be active at the same time, resulting in significantly greater residual norm growth. We added an additional regularization on the norm of the residual, which delayed the blowup significantly, but it still happened at around 20% of the training completed. Until that point its behavior was as we predicted: naive UT with PeriLN outperformed naive UT without PeriLN (using PreLN instead). We are happy to finalize these experiments, search for a stable solution, and add them to the final version of the paper, but since naive UT is a completely novel setting that we have not examined so far, unfortunately, the remaining time is not enough for us to do it during the rebuttal.
> > >
> > > We would like to thank the reviewer again for his efforts to improve the quality of our paper and for his thoughtful suggestions!

---

### Official Review · Reviewer_eZCx · 2024-07-12

**Soundness:** 3
**Presentation:** 3
**Contribution:** 3
**Rating:** 7
**Confidence:** 4

**Summary:**

This paper focuses on the problem of inefficient parameter-computation ratio in Universal Transformers (UT). UT shares parameters across layers but reduces the parameter count significantly. One naive approach is to scaling up the layer size, however, cannot be easily achieved due to the prohibitive computational resource requirements. MoEUT, the proposed approach, exploits MoE architecture in both attention and feed forward layers to address this issue. To achieve the similar performance as the standard Transformers, the authors first introduce layer grouping, which allows non-shared weights among the layers within a group. Second, they remove the layernorm in the main data path in standard Transformer to avoid gradient flow and add layernormaround the residual connections. Experiments shows MoEUT outperforms standard Transformers slightly while incurring less compute and memory costs.

**Strengths:**

1. The idea of exploiting MoE to address the parameter-compute ration in Universal Transformers is interesting. MoE itself incorporates ideas of expert sharing among similar tokens. It is an intuitively reasonable combination.
2. Sound explanations of the rationale of designing layer grouping and peri-layernorm are presented in the paper together with cited works to further support the claims.
3. The evaluation and analysis are comprehensive with deep dive to the design components.

**Weaknesses:**

1. The claim of using significantly less compute and memory should be further supported by evaluation numbers. Though Table 4 have presented detailed training hardware information of the experiments reported in the paper, it is very hard to determine the quantitative computation and memory costs. A general number like GPU hours, GPU duty cycles, DRAM utilization or costs in terms of dollars would be a much better indicator rather than existing presentation.
2. In Section 2.1, it is mentioned the load balancing loss is now considered for each sequence due to the loss explosion issue when applying on the entire batch. I may not fully understand the difference here and why full batch balancing loss would cause explosion while sequence-level won't. Further elaboration or evaluation could be helpful in explaining this.

**Questions:**

It seems the number of experts is very large as a default setting in the evaluation. What if the total number of experts is just, let's say, 8 or 16 and k is 1 or 2, similar like the sparsely-activated MoE, how the performance would change?

**Limitations:**

Limitations are addressed in the paper and there is no negative societal impact of this work.

---

> ### Author Rebuttal · Authors · 2024-08-05
>
> We would like to thank the reviewer for the valuable review and for many positive comments on the methodology of our paper. Please find our responses as follows:
>
> > The claim of using significantly less compute and memory should be further supported by evaluation numbers. Though Table 4 have presented detailed training hardware information of the experiments reported in the paper, it is very hard to determine the quantitative computation and memory costs.
>
> We will update Table 4 in the final version if the paper is accepted, to include the total GPU-hours (N_GPU * duration in the current table) for better readability. Comparing memory this way is nontrivial, as it does not scale linearly with the number of GPUs, as we are using simple data-parallel training. However, here are some direct comparisons on identical hardware for the 244M parameter models that allow directly comparing them on identical hardware, showing that MoEUT is much faster and uses less memory than the alternative UT variants:
>
> | Model | Time / training step | Memory usage/GPU |
> | ---- | ---- | ---- |
> | Dense |  443 ms/iter | 9.2 Gb |
> | Naive UT |  3559 ms/iter | 25.9 Gb |
> | MoEUT | 772 ms/iter | 9.0 Gb |
> | SUT |  1344 ms/iter  | 23.4 Gb |
>
> We measured the training iteration time and memory usage on 8 V100 32Gb GPUs. Here one “iteration” corresponds to an effective batch size of 64x1024 tokens for all models. The training iteration time was measured by using a batch size for each model that fits GPUs; models require either 1 or 2 gradient accumulation steps to achieve the effective batch size depending on their memory requirement. We measured the training time right after initialization and a warmup period. The memory usage is measured using 2 grad accumulation steps for all models for a fair comparison. Note that around ~3Gb of memory is used by the model parameters and optimizer state on each GPU.
>
> > I may not fully understand the difference here and why full batch balancing loss would cause explosion while sequence-level won't.
>
> We cannot provide a mathematical proof here, but we do have one hypothetical but intuitive explanation. If the balancing is done at the batch level, the experts can specialize in different *sequence types*. Since the batches are sampled randomly, the “variance” of utilization rate between the active experts used in different forward passes might vary highly. Some experts may specialize in *rare* sequences, thus they will be rarely trained, resulting in a “slow expert collapse”. When they are reactivated at a much later stage of training, they may cause sudden loss explosion as their training “stage” lags behind that of other more frequently used experts. On the other hand, if the balancing is done on a sequence level, the model is encouraged to use various experts in each sequence, eliminating some of this variance. This still allows some expert specialization, because the balancing is relatively weak.
>
> Using a significantly larger batch size would probably also stabilize the training, but it is prohibitively expensive with our resource budget. It seems like a useful next step for better-resourced teams, and so we will emphasize this in our next version.
>
>
>
> > What if the total number of experts is just, let's say, 8 or 16 and k is 1 or 2, similar like the sparsely-activated MoE, how the performance would change?
>
> This is an excellent question. In our experiments, we found K=1 performed significantly worse than K > 1. In order to match the MACs of the dense model, the number of experts should be at least K*N_layers, otherwise the activated experts would be “wider” than the FFN of the baseline. Furthermore, to keep the number of parameters constant, d_expert should be significantly increased.
>
> We study the effect of increasing expert size (d_expert) in Fig. 13 in the appendix, while keeping K constant. Increasing the expert size is detrimental. Decreasing K would have an additional negative effect. We analyze this independently of K in Fig 14.
>
> If the reviewer thinks additional experiments could add value to the paper, we can run an experiment with some of our MoEUT methods in a max-d_expert configuration and K=2 to see the effects of pushing these parameters to the extreme.
>
> These questions are extremely valuable for us and will enable us to improve our paper. Thank you very much.

---

> > ### Comment · Reviewer_eZCx · 2024-08-12
> >
> > Thank you very much for addressing my concerns. They are really helpful. I will maintain my current rating.

---

> > > ### Author Response · Authors · 2024-08-13
> > >
> > > Thank you very much for your response! We are glad to hear that the reviewer found our response useful!

---

### Official Review · Reviewer_83mA · 2024-07-18

**Soundness:** 3
**Presentation:** 4
**Contribution:** 3
**Rating:** 7
**Confidence:** 5

**Summary:**

The paper suggests to use Sparse MoEs together with Universal Transformers to overcome the parameter-count limitation that the latter have when parameters are shared over consecutive layers. In particular, the work suggests to use $\sigma$-MoEs (that use sigmoid activation function in the router, rather than the more popular softmax activation), and use fine-grained experts (selecting many small experts over a large pool, rather than one or two large ones over a smaller pool). The goal is to make Universal Transformers competitive on language modeling tasks, where they haven’t excelled in the past (presumably due to the smaller parameter count, compared to dense models of the same time complexity).

Two important novelties that are introduced are Layer Grouping in UTs (this allows to chose the degree of parameter sharing across layers in the UT), and relocating the LayerNorm operations (called PeriLayerNorm, as opposed to pre and post LayerNorm).

The paper includes experiments on several pretraining tasks (C4, peS2o, SlimPajama) and zero-shot evaluation on several downstream tasks. In both cases, the results achieved by MoEUT are competitive with a standard dense transformer with the same number of parameters, and often slightly better.

**Strengths:**

- The presentation of the work is excellent. From the motivation, to the explanation of the proposed method, related works, and the description of the experiments.
- The main experiments show that the goal of making UTs competitive with standard dense transformers in language modelling tasks was achieved.
- Many additional ablation experiments were conducted to try to explain how experts are selected across different layers, and tokens.
- An effort was put on implementing reasonable baselines such as $\sigma$-MoEs and Sparse UTs.
- The paper clearly states current limitations (e.g. not the most efficient implementation, which results in experiments that are 2x slower than the dense counterpart).

**Weaknesses:**

- The proposed method matches dense transformers on language modeling, but it’s barely better. This begs the question: why using MoEUTs then, rather than the simpler baseline? The answer could be better MAC efficiency at the same memory cost, but this is not clearly represented in Table 1 (and as the authors point out, it’s actually slower due to implementation limitations).
- The perplexity reported for Sparse UTs is alarmingly high. The paper mentions that proper hyperparameter tuning was done, but I’m wondering if the authors reached out to the SUT authors to make sure that everything was implemented correctly.
- A similar thing happens with $\sigma$-MoEs, which performs worse than the dense baseline when matching the parameter count, and most troubling barely better when matching MACs.


-------
Edit after rebuttal: The authors addressed most of my concerns in the rebuttal. Given their response to my and to other reviewers' comments, I'm increasing my score to "Accept".

**Questions:**

- In figure 10 and 11, do the expert indices correspond to the experts in the attention or in the feed-forward part of the block? Do the trends differ across the two?

**Limitations:**

No negative societal impact particular to this work, in my opinion.

---

> ### Author Rebuttal · Authors · 2024-08-05
>
> We would like to thank the reviewer for their insightful review and for the positive comments on the clarity and methodology of the paper. Please find our responses as follows:
>
> > The proposed method matches dense transformers on language modeling, but it’s barely better. This begs the question: why using MoEUTs then, rather than the simpler baseline?
>
> Our goal is to pave the way towards a general model that can be used for LM but is also good at systematic generalization, eventually arriving at foundation models that are more trustworthy and data efficient. In this paper, we address a single step toward that goal: we present a way to overcome the long-standing efficiency limitations of UTs (which are known to generalize better than standard transformers), which have prevented their scaling to sizes required for LLMs.
>
> > it’s actually slower due to implementation limitations
>
> It is true that our current proof-of-concept kernel is slower than a standard transformer despite requiring fewer flops, but we believe this can be mostly mitigated by better kernels. Even with the current kernels, our method is significantly faster than a naive UT or the next best UT baseline, SUT, and for most of our experiments naive UTs are not viable to run because of their memory usage and speed (see time/memory usage measurements below). Rather than replacing Transformers, we are advocating for more research on Universal Transformers with our method for larger scales.
>
> Note additionally that we investigated these models in a parameter-matched setting because we were interested in comparing their expressiveness. The number of parameters in our MoEUT can be increased very cheaply (as with other MoEs), and it is possible to achieve significantly better perplexity for marginally higher compute, potentially justifying the wall-clock time slowdown even with the current kernels. (We use a parameter-matched setup instead of this because this would give an unfair advantage to our MoEUTs compared to the baselines).
>
> Speed/memory usage of different models, showing that MoEUT is much faster compared to naive UT and SUT and uses less memory:
> | Model | Time / training step | Memory usage/GPU |
> | ---- | ---- | ---- |
> | Dense |  443 ms/iter | 9.2 Gb |
> | Naive UT |  3559 ms/iter | 25.9 Gb |
> | MoEUT | 772 ms/iter | 9.0 Gb |
> | SUT |  1344 ms/iter   | 23.4 Gb |
>
> We measured the training iteration time and memory usage on 8 V100 32Gb GPUs. Here one “iteration” corresponds to an effective batch size of 64x1024 tokens for all models. The training iteration time was measured by using a batch size for each model that fits GPUs; models require either 1 or 2 gradient accumulation steps to achieve the effective batch size depending on their memory requirement. We measured the training time right after initialization and a warmup period. The memory usage is measured using 2 grad accumulation steps for all models for a fair comparison. Note that around ~3Gb of memory is used by the model parameters and optimizer state on each GPU.
>
> > The perplexity reported for Sparse UTs is alarmingly high
>
> We confirmed via personal communication through a colleague that SUT does not work well for LM. We used the official code of the authors, with minimal modification required to adapt it to our codebase. Additionally, we conducted some ablations on the SUT models, and we found that the main cause of the bad performance is the ACT used by their model (which was presented as a fundamental building block of SUT). By disabling ACT, the gap between MoEUT and SUT is much smaller, while MoEUT remains consistently better:
>
> | Dataset | Model Size | Baseline | MoEUT | SUT w.o. ACT | SUT |
> | --- | --- | --- | --- | --- | --- |
> |C4| 44M | 18.9 | 18.2 | 21.5 | 40.5 |
> | | 244 M | 13.3 | 13.2 | 14.5 | 20.0 |
> | peS2o |  44M | 11.5 | 11.1 | 12.7 | 25.0 |
> | | 244M | 8.6 | 8.5 | 9.3 | 20.4|
>
> > A similar thing happens with 𝜎-MoEs, which performs worse than the dense baseline when matching the parameter count.
>
> Please note that our baselines are much stronger than those reported in the 𝜎-MoE paper: on C4, we achieve a perplexity of 13.4 using 244M parameters vs. 17.79 reported by the 𝜎-MoE paper using 266M.
>
> This difference comes from two modifications. First, the 𝜎-MoE paper follows the experimental protocol of Transformer XL: we used their official 𝜎-MoE codebase, but improved their baseline by using RoPE and no XL cache. Second, they use dropout in the FFN layers of their baseline, and ‘expert dropout’ in the 𝜎-MoE. Here we disabled all dropouts in all our models as we use sub-epoch training. This resulted in perplexity improvements with a higher gain for the baseline than for 𝜎-MoE.
>
> > In figure 10 and 11, do the expert indices correspond to the experts in the attention or in the feed-forward part of the block?
>
> Those expert indices correspond to the feed-forward part. We will clarify this in the final version. Thank you for pointing this out!
>
> We believe our response above resolves all the concerns that the reviewer has raised. If the reviewer finds our response useful, please consider increasing the score. Thank you very much.

---

> > ### Comment · Reviewer_83mA · 2024-08-13
> >
> > I thank the authors very much for their detailed response to my comments, and the other reviewers' comments as well.
> >
> > In particular, I appreciate very much the fact that the authors made the effort to fairly reproduce the baselines, and contacted the authors to make sure that they were correctly represented.
> >
> > Thus, given that I feel that my concerns have mostly been addressed, I will increase the score and recommend the acceptance of the paper.

---

> > > ### Author Response · Authors · 2024-08-14
> > >
> > > Thank you very much for the increased score! We are glad to hear that the reviewer found our response useful! Thank you again for your valuable feedback!

---

### Comment · Area_Chair_zoLu · 2024-08-12
**please respond to rebuttal**

Dear reviewers,

The author-reviewer discussion period will end soon. Please make sure you read the authors' rebuttal and respond to it. If you have additional questions after reading the rebuttal please discuss with the authors. For those who have done so, thank you!

AC

---

### Decision · Program_Chairs · 2024-09-25

**Decision:**

Accept (poster)

**Comment:**

This paper introduces mixture-of-experts (MoE) in the universal Transformer (UT) architecture with parameter sharing across layers.  UTs have a limitation on parameter-compute ratio.  This parameter count problem makes UTs disadvantageous when dealing with parameter-dominant tasks such as modern language modeling.   Specifically, the authors introduce MoE in both attention and feed-forward layers of UT.  In addition, the authors propose two techniques in the so-called MoEUT. One is layer grouping and the other is to relocate the layer normalization. Experiments shows MoEUT outperforms standard Transformers slightly while incurring less compute and memory costs.  In the rebuttal and follow-up discussions, the authors have addressed most of the concerns by the reviewers with additional experimental results and elaborated explanation. Overall this is an interesting paper and all reviewers recommend acceptance.  That being said, the performance of the proposed MoEUT trained on large scale data and models is still yet to to seen and further technical improvements may need to be investigated.